# Lack of protective effect of chloroquine derivatives on COVID-19 disease in a Spanish sample of chronically treated patients

**Marina Laplana**[1], **Oriol Yuguero**[2,3], **Joan Fibla**[3,4]*

**1** Departament de Ciència Animal, Universitat de Lleida, Lleida, Spain, **2** Hospital Universitari Arnau de Vilanova de Lleida, Lleida, Spain, **3** Institut de Recerca Biomedica de Lleida, Lleida, Spain, **4** Departament de Ciències Mèdiques Bàsiques, Unitat de Genètica Humana, Lleida, Spain

* joan.fibla@udl.cat

## Abstract

### Background

The search for a SARS-CoV-2 treatment has emerged as a worldwide priority. We evaluated the role of chloroquine and its derivatives in COVID-19 in Spanish individuals.

### Methods

We performed a survey addressed to patients regularly taking chloroquine and its derivatives for the control of their autoimmune diseases. The survey was distributed with special attention to Spanish patient associations centred on autoimmune diseases and rheumatology and to the general population. A sample of untreated subjects was matched to the treated group according to sex, age range and incidence region. COVID-19 disease prevalence was compared between treated and untreated-matched control sample.

### Results

A total of 319 surveys of patients regularly taking chloroquine and its derivatives were recovered for further analysis. The prevalence of declared COVID-19 status in the treated group was 5.3% and the mean prevalence among the untreated-matched groups was 3.4%. A community exposition to COVID-19 was associated with a greater prevalence of COVID-19 in both, treated (17.0% vs. 3.2%; p-value<0.001) and untreated groups (13.4% vs. 1.1%; p-value = 0.027).

### Conclusion

We did not find differences of reported COVID-19 cases between treated and untreated groups, indicating a lack of protection by regular administration of chloroquine and its derivative drugs on COVID-19 infection. Of relevance, data indicates that patients that regularly take chloroquine derivatives are exposed to SARS-CoV-2 infection and must take the same protection measures as the general population.

**Data Availability Statement:** All relevant data are within the manuscript and its Supporting Information files.

**Funding:** The author(s) received no specific funding for this work.

**Competing interests:** The authors declare that the research was conducted in the absence of any commercial or financial relationships that could be construed as a potential conflict of interest. This does not alter our adherence to PLOS ONE policies on sharing data and materials.

## Introduction

The search for an effective treatment against SARS-CoV-2 infection has emerged as a world-wide priority. A promising strategy to fight the virus causing the newest pandemic is the identification of already available drugs active against other diseases that can be effective also against this new SARS-CoV-2 infection [1].

We present here an evaluation of the potential protective effect of chloroquine and its derived drugs on the prevalence of COVID-19 in patients undergoing an active treatment with these drugs.

Chloroquine and its derived drugs such as hydroxychloroquine have shown *in vitro* and *in vivo* effectiveness against viral infections such as SARS [2], influenza A H5N1, and Zika. Chloroquine is believed to interfere with both the entry and exit of viruses from cells hosts, as well as in the manifestation of acute respiratory syndrome. The virus enters the cells through binding to the angiotensin-converting enzyme 2 (ACE2). Chloroquine can reduce ACE2 glycosylation, thereby preventing viruses from effectively binding to cells [3]. On the other hand, chloroquine accumulates in lysosomes, increasing the endosome pH levels which interferes with the viral particle release process [4]. In addition, chloroquine could block the production of proinflammatory cytokines, thus preventing the pathway that subsequently leads to acute respiratory syndrome [5]. Two recent clinical trials have presented inconclusive evidence on the effectiveness of chloroquine treatment in COVID-19 disease in Chinese [6] and French [7] populations. Both studies have supported the use of chloroquine or chloroquine derivatives against COVID-19, however, the design and conclusions of both studies have been questioned [8].

Chloroquine and its derivatives such as chloroquine phosphate or hydroxychloroquine are commonly used in the treatment of autoimmune diseases. Not without serious side effects, the use of these drugs under medical prescription is widely spread. It has been proposed hydroxychloroquine as a prophylaxis treatment against SARS-CoV-2 infection for exposed caregivers [9–11]. Our hypothesis is that if chloroquine treatment is effective against SARS-CoV-2 infection, those patients following an active chloroquine or derivative drug treatment would be protected against the infection or against COVID-19 adverse effects. Thus, our study aims to test this hypothesis by evaluating the incidence of COVID-19 disease in the population according to chloroquine treatment subgroups through a survey.

## Material and methods

### Survey design and data collection

A survey was distributed in Spanish and Catalan languages and conceived to be conducted electronically via smartphone or personal computer, and therefore it was also designed to ensure accessibility and simplicity to facilitate its completion. A copy of the original survey in Spanish and Catalan languages and a translated English version can be found as S1–S3 Texts. Information about the project and a link to the URL of the survey were disseminated in the press and via social media and email, with special attention to Spanish patient associations centered on autoimmune diseases and rheumatology. The survey included demographic questions about gender, age range, and Autonomous Community of residence, as well as questions pertaining to health-status outcomes such as treatment, COVID-19 diagnosis and symptoms due to COVID-19 infection. In addition, questions about infection diagnosis and symptoms in close relatives and friends were also included. Because the first cases of COVID-19 were reported in Spain in March 2020, our survey collects cases that occurred between March and May 2020.

In order to evaluate the effect of chloroquine and its derivatives on COVID-19 infection risk, we have taken advantage of the unique available source of people receiving this drug as a chronic treatment, the group of patients with autoimmune diseases. However, at time of recruitment a very low percentage of COVID-19 patients had an autoimmune disease (less than 1%) [12], while age, gender and region of origin were the main known factors affecting SARS-CoV-2 infection in Spain's autonomies. In addition, the recruitment of untreated cases had to comprise a large number of individuals to be able to do the proper matching between treated and untreated individuals, being the treated patients the limiting group. According to this, we decided to use individuals from the general population as untreated cases considering age, gender and Autonomous Community of residence as the main selection criteria for recruitment.

Individuals of any age above legal age (18 years old) with residence in Spain were eligible for inclusion in the study. Individuals undergoing a stable chloroquine or derived drug treatment before the COVID-19 pandemic were classified as treated (treated group) while individuals without treatment and those beginning acute treatment in the last three months (January-March) were classified as untreated (untreated group). Individuals beginning acute treatment were receiving it as a treatment after being infected with SARS-CoV-2, thus, they were not under treatment prior to infection.

The Clinical Research Ethics Committee of the Hospital Universitari Arnau de Vilanova in Lleida approved the study (Ref: CEIC-2257).

## Statistical analysis

Statistical analyses were performed with R software (v3.6.0) and IBM SPSS v21 (IBM corporation, NY, USA). Prior to the analysis, reported autonomy of residence in Spain were grouped according to COVID-19 incidence as stated in June 1 by the Spanish Ministry of Health [13] as follows: Incidence region 1 (incidence<200/100,000 inhabitants) that includes Andalucía, Canarias, Ceuta, Illes Balears, Melilla and Región de Murcia; Incidence region 2 (incidence 200-500/100,000 inhabitants) that includes Aragón, Cantabria, Comunidad Valenciana, Extremadura, Galicia and Principado de Asturias; Incidence region 3 (incidence 500–1,000/100,000 inhabitants) that includes Castilla y León, Castilla-La Mancha, Catalunya, Euskadi and Navarra; and Incidence region 4 (incidence>1,000/100,000 inhabitants) that includes Comunidad de Madrid and La Rioja. Age categories were assigned according to the following age ranges: 18–50, 51–65 and >65 years old. Data on symptoms related to COVID-19 infection were collected and used to assign individuals as suspected COVID-19 cases when reporting loss of taste or smell and/or three or more COVID-19 associated symptoms [14].

A sample of untreated subjects was matched to the treated group according to sex, age range and incidence region with e1071 R package. The matching process was repeated using a bootstrap strategy and re-sampling of the untreated-matched dataset was repeated 1,000 times to obtain the distribution and mean values of the descriptive statistics such as age range, gender, incidence region and declared COVID-19 prevalence. Statistical analysis was performed to validate the appropriate distribution of the demographic characteristics within each group what rules out the possibility of a bias towards one of the variables. In addition, tests between treated and untreated groups were used to confirm the proper matching of the subjects. Comparisons were performed by Fisher exact test. P-value <0.05 was considered statistically significant. Comparison of differences among treated group and 1000 replicates of untreated-matched groups was performed by Fisher exact test. In addition, the difference between the two proportions and a 95% confidence interval for this difference was performed using the comparison of proportions method (Chi-squared test).

## Results

Overall, 2295 individuals completed the surveys between May and June 2020. From these, data collection was complete, with all key data for the study, for 2161 individuals (94.2%). These completed surveys were checked for inconsistencies and possible duplicated values as well as to fulfil the eligibility criteria, 11 entries were excluded from further analysis. The final number of completed surveys used for the study was 2150. A copy of the anonymized data set is provided in S1 Data. Among them, 319 (14.8%) were from patients following an active chloroquine or derived drug treatment and have been included in the treatment group, and 1831 (85.2%) have been included in the untreated group and serves as the source to obtain the untreated-matched subgroups. We note that 94% of the treatment group individuals were following a hydroxychloroquine treatment.

The main descriptive characteristics of both treated (n = 319) and untreated-matched (n = 319) subgroups, arranged according to their declared COVID-19 status, are reported in Table 1. Distribution of declared COVID-19 status did not differ significantly within age group, gender and incidence region (Table 1). In contrast, having a community exposition

**Table 1. Demographic characteristics of persons taking chloroquine or derivatives (treated) and matched control sample (untreated) according to their declared COVID-19 status.**

| | Treated (n = 319) | | | | Untreated (n[a] = 319) | | | | Treated vs Untreated |
|---|---|---|---|---|---|---|---|---|---|
| | COVID-19 (+) | COVID-19 (-) | Totals | P-value | COVID-19 (+) | COVID-19 (-) | Totals | P-value | P-value |
| Age range, n (%) | | | | | | | | | |
| 18–50 | 9 (4.7) | 182 (95.3) | 191 | 0.433 | 6.7 (3.5) | 185.4 (96.5) | 192.2 | 0.999 | 0.981 |
| 51–65 | 8 (7.1) | 105 (92.9) | 113 | | 3.9 (3.5) | 107.5 (96.5) | 111.3 | | |
| > 65 | 0 (0.0) | 15 (100.0) | 15 | | 0.3 (2.1) | 15.2 (97.9) | 15.5 | | |
| Total | 17 (5.3) | 302 (94.7) | 319 | | 10.9 (3.4) | 308.1 (96.6) | 319 | | |
| Sex, n (%) | | | | | | | | | |
| Male | 1 (4.8) | 20 (95.2) | 21 | 0.905 | 1.1 (5.0) | 20.2 (95.0) | 21.2 | 0.973 | 1.000 |
| Female | 16 (5.4) | 282 (94.6) | 298 | | 9.8 (3.3) | 287.9 (96.7) | 297.8 | | |
| Total | 17 (5.3) | 302 (94.7) | 319 | | 10.9 (3.4) | 308.1 (96.6) | 319 | | |
| Incidence region[b], n (%) | | | | | | | | | |
| R1 | 1 (2.4) | 40 (97.6) | 41 | 0.420 | 0.0 (0.0) | 41.1 (100.0) | 41.1 | 0.275 | 0.215 |
| R2 | 1 (2.8) | 35 (97.2) | 36 | | 1.0 (3.4) | 28.7 (96.6) | 29.7 | | |
| R3 | 11 (5.4) | 193 (94.6) | 204 | | 6.1 (2.7) | 218.8 (97.3) | 225.0 | | |
| R4 | 4 (11.4) | 31 (88.6) | 35 | | 3.2 (15.9) | 17.0 (84.1) | 20.2 | | |
| Unknown | 0 (0.0) | 3 (100.0) | 3 | | 0.5 (17.3) | 2.5 (82.7) | 3.0 | | |
| Total | 17 (5.3) | 302 (94.7) | 319 | | 10.9 (3.4) | 308.1 (96.6) | 319 | | |
| Community exposition, n (%) | | | | | | | | | |
| Exposed[c] | 8 (17.0) | 39 (83.0) | 47 | <0.001 | 8.2 (13.3) | 53.3 (86.7) | 61.5 | 0.027 | 0.300 |
| Unexposed | 7 (3.2) | 210 (96.8) | 217 | | 2.2 (1.1) | 204.4 (98.9) | 206.6 | | |
| Unknown | 2 (3.6) | 53 (96.4) | 55 | | 0.5 (1.0) | 50.4 (99.0) | 50.9 | | |
| Total | 17 (5.3) | 302 (94.7) | 319 | | 10.9 (3.4) | 308.1 (96.6) | 319 | | |

COVID-19: disease caused by SARS-CoV-2 infection.

[a] Mean after 1000 replicates of matched untreated control samples.

[b] Regions grouped by incidence of COVID-19. IR1 (incidence <200/100,000 inhabitants) that includes Andalucía, Canarias, Ceuta, Illes Balears, Melilla and Región de Murcia; IR2 (incidence 200-500/100,000 inhabitants) that includes Aragón, Cantabria, Comunidad Valenciana, Extremadura, Galicia and Principado de Asturias; IR3 (incidence 500–1,000/100,000 inhabitants) that includes Castilla y León, Castilla-La Mancha, Catalunya, Euskadi and Navarra; and IR4 (incidence >1,000/100,000 inhabitants) that includes Comunidad de Madrid and La Rioja.

[c] Exposed were those individuals declaring a COVID-19 positive case in a close family member or flatmate.

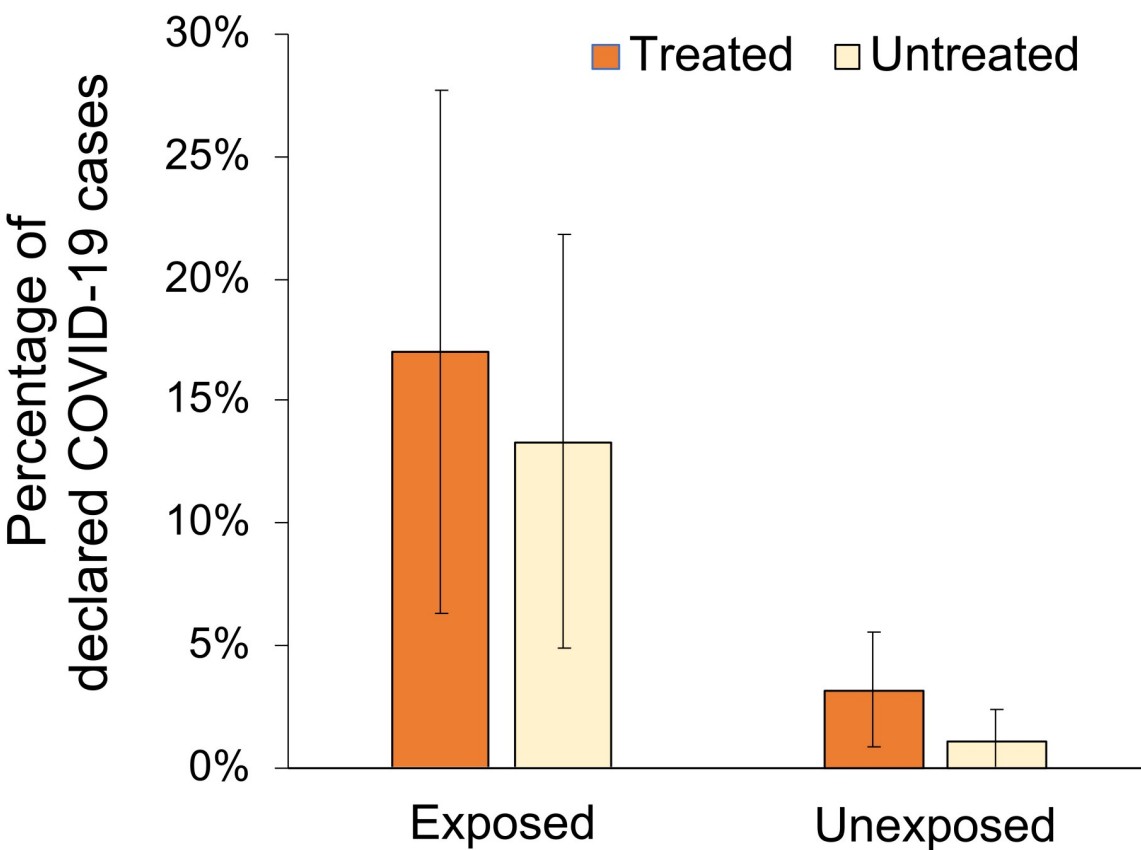

**Fig 1. Percentage of declared COVID-19 cases according to treatment status and community exposition.** Treated group refers to patients regularly taking chloroquine and its derivatives. Exposed are those individuals declaring a COVID-19 positive case in a close family member or flatmate. Error bars depict the 95% confidence interval.

(defined as those individuals declaring a COVID-19 positive case in a close family member or flatmate) was associated with a greater prevalence of COVID-19 disease when compared with non-exposed individuals in both treated (17.0%, 95%CI 12.9%-21.1% vs. 3.2%, 95%CI 1.3%-5.2%; p-value<0.001) and untreated subgroups (13.4%, 95%CI 9.6%-17.1% vs. 1.1%, 95%CI 0%-2.2%; p-value = 0.027) (Table 1 and Fig 1). Furthermore, we evaluated the distribution of the descriptive characteristics between treated and untreated groups denoting no statistical significant differences, and thus, confirming the appropriate matching of the groups (Table 1).

The prevalence of declared COVID-19 status in the treated group was 5.3% (95%CI 2.9–7.8) and the mean prevalence among the untreated-matched groups was 3.4% (95%CI 1.4–5.4). Testing differences among treated and 1000 replicates of untreated-matched groups reveals significant differences only in 28 comparisons (P = 0.972). In addition, the difference of proportions of declared COVID-19 cases between both groups did not reach statistical significance (difference 1.9%, 95%CI: 0–5.3; P = 0.240). Furthermore, the prevalence of suspected COVID-19 patients in treated subjects was of 18.8% (95%CI 14.5–23.1) and the mean prevalence among the untreated-matched groups was 15.7% (95%CI 11.7–19.7). Neither the comparison of the prevalence nor the distribution of the difference of declared COVID-19 cases among groups showed significant differences. These figures are nearly similar to those recently found in the study of the seroprevalence of IgG antibodies against SARS-CoV-2 in the Spanish population showing an estimated prevalence of 5% (95%CI 4.7%-5.4%), and a prevalence of suspected COVID-19 cases of nearly 20% [14].

## Discussion

Our results show no differences in COVID-19 prevalence among untreated and chronically treated individuals with chloroquine or derivative drugs. Independently of the exposure, both groups showed the same prevalence of COVID-19 disease or suspected COVID-19 disease according to symptoms. We must note that we found a clear association between the COVID-19 disease prevalence and exposure to a close family member or flatmate positive for COVID-19 in both, treated and untreated subjects, that points to a lack of any protective effect on SARS-CoV-2 infection attributable to chronic treatment with chloroquine or derivative drugs.

We should mention some limitations of our study such as a limited power to detect small changes in prevalence between treatment groups. However, the design of this study addressed the need to collect and analyse data within a particularly short period of time due to the rapid onset and progression of the pandemic, as well as the urgency of identifying and evaluating rapidly possible therapies, thus partially compensating the reduced sample size. The prevalence of COVID-19 found in our study is similar to the seroprevalence of IgG antibodies against SARS-CoV-2 in the Spanish population [14], which is higher than the reported COVID-19 prevalence in the general population based on RNA's virus detection [13]. This difference could be attributed to self-reported disease and the diagnosis of COVID-19 by medical practitioners, which in many cases does not involve results of diagnostic tests due to the lack of such tests. Finally, we could not eliminate completely the possibility of some bias due to the intrinsic condition of the individuals within the treatment group that are undergoing chloroquine or derivative drug treatment due to other diseases that alter their health status and may have different comorbidities. A previous study reported a low portion of autoimmune diseases patients as COVID-19 cases (less than 1%) [12]. On the other hand, the main factors reported to affect SARS-CoV-2 infection were age, gender and region of origin in Spain's autonomies. Despite the lack of evidence that autoimmune diseases affect the risk of infection, we cannot completely discard a possible bias caused by the autoimmune status of cases. However, we assume a negligible effect caused by this possible bias in comparison with the other matching parameters such as sex, age and place of residence, that have allowed us to obtain a large and representative N of the different population groups. Finally, we lack information about chloroquine or hydroxychloroquine treatment doses for each subject what, in case of low doses could not be enough to show an effect in the prevention of COVID-19. However, based on the Spanish Agency for Medicines and Health Products, the standard treatment for autoimmune diseases such as lupus or rheumatoid arthritis usually ranges from 200 to 600mg of hydroxychloroquine per day, what is in line with the dose given as a treatment to COVID-19 patients. Thus, the doses taken regularly by the subjects of the study and the doses used as treatment for infected patients should not differ significantly what makes us think that this should not represent a major limitation of our study.

However, our results are in line with a recent study conducting a randomized trial that reported no effect of hydroxychloroquine when used as a post exposure prophylaxis for COVID-19 [15].

## Conclusion

All these data together point towards a lack of a protective effect of chloroquine or derivative drugs as a prophylaxis for COVID-19, including prophylactic treatment before and after exposure in patients with autoimmune diseases or other chronic conditions that require these treatments, and potentially increase the risk for SARS-CoV-2 infection per se. Of relevance, data indicates that people that regularly take chloroquine derivatives are exposed to SARS-CoV-2 infection and must take the same protection measures as the general population. These data

should be considered in the prevention and treatment protocols made by health policymakers for the management of the disease in new outbreaks. Finally, efficacy of chloroquine and its derivatives in preventing SARS-CoV-2 infection will be determined by upcoming clinical trials.

## Supporting information

**S1 Text. Survey questions in Spanish.**
(DOCX)

**S2 Text. Survey questions in Catalan.**
(DOCX)

**S3 Text. Survey questions translated to English.**
(DOCX)

**S1 Data. Anonymised data set.**
(XLSX)

## Acknowledgments

This study and the research behind it would not have been possible without the selfless contribution of all the people who participated and that took time to respond the survey. We would like to thank Dr. Serafi (Tapi) Piñol for his valuable comments and suggestions.

## Author Contributions

**Conceptualization:** Joan Fibla.

**Data curation:** Marina Laplana, Oriol Yuguero, Joan Fibla.

**Formal analysis:** Marina Laplana, Oriol Yuguero, Joan Fibla.

**Supervision:** Joan Fibla.

**Validation:** Joan Fibla.

**Writing – original draft:** Marina Laplana, Joan Fibla.

**Writing – review & editing:** Marina Laplana, Oriol Yuguero, Joan Fibla.

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
