## [Editor Report · Decision Letter 0]

4 Sep 2020

PONE-D-20-26952

Lack of protective effect of chloroquine derivatives on COVID-19 disease in a Spanish sample of chronically treated patients.

PLOS ONE

Dear Dr. Fibla,

Thank you for submitting your manuscript to PLOS ONE. After initial evaluation, we feel that authors shall address this following comment from the editor prior to an external review.

This is a very valuable and interesting manuscript that attempts to address the most urgent and heavily debated question, e.g. whether chloroquine is effective on COVID-19 or not. One major concern is the untreatment group which should be patients with autoimmune diseases without receiving chloroquine treatment rather than general population. Are patients with autoimmune diseases more vulnerable to be infected as compared with general population?  

Therefore, we invite you to submit a revised version of the manuscript that addresses this point.

We look forward to receiving your revised manuscript.

Kind regards,

Wenbin Tan

Academic Editor

PLOS ONE

2. Please include additional information regarding the survey used in the study and ensure that you have provided sufficient details that others could replicate the analyses. For instance, if you developed the survey as part of this study and it is not under a copyright more restrictive than CC-BY, please include a copy, in both the original language and English, as Supporting Information.

In addition, please indicate whether the survey was validated.

---

## [Author Response · Author response to Decision Letter 0]

18 Sep 2020

Response to Academic Editor,

Dear Dr. Tan,

We greatly appreciate your comments and suggestions in relation to our submitted manuscript “Lack of protective effect of chloroquine derivatives on COVID-19 disease in a Spanish sample of chronically treated patients” (PONE-D-20-26952). 

As you point out, we address one of the most urgent and heavily debated questions regarding intervention in the COVID-19 pandemic, namely (i.e.) whether or not chloroquine is effective on COVID-19. This question arose during the first month of the pandemic in response to controversial publications supporting a role of chloroquine derivatives, in special hydroxychloroquine, in protecting against COVID-19 disease. In fact, the Food and Drug Administration (FDA), via its Emergency Use Authorization (EUA) authority, on March 28th provided authorization for use of hydroxychloroquine for treating patients with COVID-19, and it was not until June that this authorization was revoked. It was in this controversial context in which, at the end of March, we designed our survey in order to address whether chloroquine is effective against COVID-19. 

In order to do so, we took advantage of the only available source of people receiving this drug chronically as a treatment, the group of patients with autoimmune diseases. Thus, our main objective was to know if the prevalence of declared COVID-19 disease in this group was different from that in people that do not receive this drug.

As important as the selection criteria were for inclusion in the treated group, also were the selection criteria for inclusion in the reference group. At the moment of designing the survey, in terms of the risk of infection, what had been observed was that a very low percentage of COVID-19 patients had an autoimmune disease (less than 1%) (Emmi, G., Bettiol, et al. 2020, Autoimmunity Reviews 19(7), 102575. https://dx.doi.org/10.1016/j.autrev.2020.102575), while age, gender and region of origin were the main known factors affecting SARS-CoV-2 infection in Spain's autonomies. According to this, we considered age, gender and region of origin as the main selection criteria for the untreated control group. 

In addition, the set of untreated cases had to comprise a large number of individuals in order to be able to do the proper matching between treated and untreated individuals, with the treated patients being the limiting group. This matching process would have been difficult to achieve, in a reasonable time, if our control group came only from autoimmune disease patients without treatment.

As we considered that it was a priority to arrive at an answer as quickly as possible, and taking all the available evidence into account, we chose as reference group people from the general population, in order to be able to obtain a representative matchable group that would allow us to answer our question with the maximum guarantees of reliability.

We are aware that this design has limitations. Despite the lack of evidence that autoimmune diseases affect the risk of infection, we cannot completely discard a possible bias caused by the autoimmune status of cases. However, we assume a negligible effect caused by this possible bias in comparison with the other matching parameters such as sex, age and place of residence, that have allowed us to obtain a large and representative N of the different population groups. For these reasons, we consider that the selection of the general population as a reference group is appropriate and allows us to answer, with enough rigor, the main question of this project.

We have included some of these reflections in the justification of the inclusion criteria, as well as in the discussion section of the manuscript that we hope will allow readers to understand more precisely the scope and limitations of the work here presented.

Moreover, we have included a copy of the survey in Spanish, Catalan and an English translation and the minimal anonymized data set necessary to ensure replicability of our findings as Supporting Information. Furthermore, we confirm that the survey was validated and checked for inconsistencies.

Taking into account all the aforementioned considerations, we kindly ask you to consider this amended version of the manuscript for publication in Plos One.

Yours Sincerely,

Dr. Joan Fibla

Departament de Ciències Mèdiques Bàsiques

Universitat de Lleida-IRBLleida

Edifici Biomedicina I 

Av. Rovira Roure, 80 

25198 Lleida, Catalonia, Spain

---

## [Decision Letter · Decision Letter 1]

21 Oct 2020

PONE-D-20-26952R1

Lack of protective effect of chloroquine derivatives on COVID-19 disease in a Spanish sample of chronically treated patients.

PLOS ONE

Dear Dr. Fibla,

Thank you for submitting your manuscript to PLOS ONE. After careful consideration, we feel that it has merit but does not fully meet PLOS ONE’s publication criteria as it currently stands. Therefore, we invite you to submit a revised version of the manuscript that addresses the points raised during the review process.

We look forward to receiving your revised manuscript.

Kind regards,

Wenbin Tan

Academic Editor

PLOS ONE

Review Comments to the Author

Reviewer #1: The article by Laplana et al. is a survey-based research article aimed to address the potential effectiveness of chloroquine and its derivatives to prevent SARS-CoV-2 infection among patients regularly taking these drugs for treatment of their autoimmune diseases. Authors found that difference in mean prevalence of declared COVID-19 status was not statistically significant between the treated group (5.3%) and the untreated-matched group (3.4%).

Minor Comments

Line 64: Authors should quote also: PMID: 32693652 and PMID: 32311322

Apart from being based on surveys, study has other limitations that should be acknowledged, such as lack of information about chloroquine/HCQ dose. Moreover, “untreated” cases also included individuals beginning acute treatment during January-March 2020.

Lines 178-184: conclusions should be rephrased, pointing out that this study may suggest lack of efficacy of chloroquine and its derivatives as a prophylactic strategy against COVID-19 in patients with autoimmune diseases or other chronic conditions which require CQ/HCQ treatment and potentially increase the risk for SARS-CoV-2 infection per se. Therefore, efficacy of chloroquine and its derivatives in preventing SARS-CoV-2 infection will be determined by upcoming clinical trials.

Reviewer #2: This paper cumulates all the conditions that make the understanding, around hydroxychloroquine and COVID-19, completely incomprehensible.

Firstly, these are not people diagnosed with COVID-19 but people interviewed by telephone. At present, recent work carried out, determining the predictive value of the clinical diagnosis or diagnosis considered by people appears to be less than 50%. Based on these elements, it is impossible to make statistics, especially with such low numbers. In practice, COVID-19 is a disease for which a biological diagnosis is made, not a diagnosis by telephone.

Furthermore, the number of people interviewed is extremely low, and finally, there is no control group, as has been done in other, much more convincing studies in the literature, of patients with a comparable pathology and taking another treatment than hydroxychloroquine.

In practice, this paper is just a paper of opportunism, which does not contribute to knowledge, and is just being swept away by the flood of COVID-19 and hydroxychlroquine delirium.

Reviewer #3: This manuscript describes an interesting study on lack of protective effect of chloroquine derivatives on COVID-19 disease among sample of chronically treated patients in Spain. The manuscript attempts to address an important global public health issue – COVID-19 disease. Understanding treatments that are effective in managing the disease is of paramount importance to policymakers and program managers in addressing this global public health issue. Presently, there is paucity of data on the subject so this manuscript if published could provide some information on the subject.

However, my only concern is wrong analysis conducted and the case-control design used in ruling out the protective effect of chloroquine derivatives on COVID-19 disease.

See more details below:

Introduction:

a) The authors provided enough background to the study.

Methods:

b) The methods employed by the authors in the analysis and presentation of the data appears problematic (see Table 1). The authors appeared to have analysed and presented the results for the treated group against the demographic characteristics separately and also for the untreated group separately. This is inappropriate because the analyses and the presentation of the results must be done across the treated and the untreated groups to be able to identify any protective effect of the intervention in the treated group as against the untreated group. This is a major flaw in this paper. This must be corrected, and more details provided in the methods section about the analysis plans.

c) Also, randomised studies design could have provided a better alternative to case control design. Thus, absence of randomization could potentially lead to unreliable parameter estimates and its associated misleading conclusion. However, the authors raised genuine concerns in their first round of revision as to why they have used the case control as oppose to other relatively better designs. This is a limitation for the study.

d) The selection of controls (untreated) was also problematic, but the authors provided justification for this in their first round of revision which I consider to be satisfactory.

e) Furthermore, the analysis did not adjust for potential confounding among demographic characteristics under consideration in this study because only bivariate analysis via Fisher’s Exact test was conducted. Multivariable analysis is needed to address this concern.

Results:

f) The results appear not to address the research question and the hypothesis postulated. Re-analysis is required. For example, if age range is not statistically significant for the treated group and also not significant for the untreated group, this could not mean that chloroquine or derivatives has no protective effect on COVID-19 disease or vice versa.

g) Also, descriptively, you might observe some differences in percentages in COVID-19 prevalence between treated and untreated group BUT statistically, that may not be important (i.e. statistical significance) or vice versa. Hence decision to conclude on any protective effect of chloroquine derivatives on COVID-19 disease should be based on statistical significance based on sound hypothesis testing.

Discussion and conclusion:

h) The issues raised in the methods and the result sections should be addressed to enable the reviewer do sound evaluation of the discussion and conclusion sections.

---

## [Author Response · Author response to Decision Letter 1]

23 Oct 2020

Response to Reviewers,

Reviewer #1: The article by Laplana et al. is a survey-based research article aimed to address the potential effectiveness of chloroquine and its derivatives to prevent SARS-CoV-2 infection among patients regularly taking these drugs for treatment of their autoimmune diseases. Authors found that difference in mean prevalence of declared COVID-19 status was not statistically significant between the treated group (5.3%) and the untreated-matched group (3.4%).

Minor Comments

Line 64: Authors should quote also: PMID: 32693652 and PMID: 32311322

Response: We appreciate the revision of the literature by the reviewer and we have incorporated these recent publications regarding our topic of interest.

Apart from being based on surveys, study has other limitations that should be acknowledged, such as lack of information about chloroquine/HCQ dose. Moreover, “untreated” cases also included individuals beginning acute treatment during January-March 2020.

Response: We welcome the comments about the limitations of the study and have addressed them in the text of the manuscript in order to clarify. The new text regarding acute treatment can be found in the materials and methods, Survey design and data collection section (lines 98-101) and the limitation on chloroquine/HCQ dose information has been included in the discussion section (lines 199-206).

“Individuals beginning acute treatment were receiving it as a treatment after being infected with SARS-CoV-2, thus, they were not under treatment prior to infection.”

“Finally, we lack information about chloroquine or hydroxychloroquine treatment doses for each subject what, in case of low doses could not be enough to show an effect in the prevention of COVID-19. However, based on the Spanish Agency for Medicines and Health Products, the standard treatment for autoimmune diseases such as lupus or rheumatoid arthritis usually ranges from 200 to 600mg of hydroxychloroquine per day, what is in line with the dose given as a treatment to COVID-19 patients. Thus, the doses taken regularly by the subjects of the study and the doses used as treatment for infected patients should not differ significantly what makes us think that this should not represent a major limitation of our study.”

Lines 178-184: conclusions should be rephrased, pointing out that this study may suggest lack of efficacy of chloroquine and its derivatives as a prophylactic strategy against COVID-19 in patients with autoimmune diseases or other chronic conditions which require CQ/HCQ treatment and potentially increase the risk for SARS-CoV-2 infection per se. Therefore, efficacy of chloroquine and its derivatives in preventing SARS-CoV-2 infection will be determined by upcoming clinical trials.

Response: We have rephrased the conclusions (lines 210-218) in order to emphasize the validity of our results in relation to patients of autoimmune diseases or other chronic conditions which require CQ/HCQ treatment and not to the general population. In addition we have included a sentence making clear the necessity of clinical trials that will experimentally confirm these results. The new text is as follows:

“All these data together point towards a lack of a protective effect of chloroquine or derivative drugs as a prophylaxis for COVID-19, including prophylactic treatment before and after exposure in patients with autoimmune diseases or other chronic conditions that require these treatments, and potentially increase the risk for SARS-CoV-2 infection per se. Of relevance, data indicates that people that regularly take chloroquine derivatives are exposed to SARS-CoV-2 infection and must take the same protection measures as the general population. These data should be considered in the prevention and treatment protocols made by health policymakers for the management of the disease in new outbreaks. Finally, we lack information about chloroquine or hydroxychloroquine treatment doses for each subject what, in case of low doses could not be enough to show an effect in the prevention of COVID-19. However, based on the Spanish Agency for Medicines and Health Products, the standard treatment for autoimmune diseases such as lupus or rheumatoid arthritis usually ranges from 200 to 600mg of hydroxychloroquine per day, what is in line with the dose given as a treatment to COVID-19 patients. Thus, the doses taken regularly by the subjects of the study and the doses used as treatment for infected patients should not differ significantly what makes us think that this should not represent a major limitation of our study.”

Reviewer #2: This paper cumulates all the conditions that make the understanding, around hydroxychloroquine and COVID-19, completely incomprehensible.

Response: We are very sorry that our study is not adequate for the reviewer. 

We will try to clarify some of the points raised in order to proof that our study was properly performed addressing one of the most urgent and heavily debated questions regarding intervention in the COVID-19 pandemic, namely (i.e.) whether or not chloroquine is effective on COVID-19. Our study was performed during the initial months of the pandemic in response to controversial publications supporting a role of chloroquine derivatives, in special hydroxychloroquine, in protecting against COVID-19 disease. At that point we considered that there was the need of providing a quick response on the prophylactic effect of chloroquine and its derivatives against SARS-CoV-2 infection and thus, we designed the epidemiological study and the survey as a tool that would be much faster than any experimental design. Of course, we acknowledge that has been a delay in publication and that now the study may seem less important but we believe that this should not be a reason to discard it from publication.

Firstly, these are not people diagnosed with COVID-19 but people interviewed by telephone. At present, recent work carried out, determining the predictive value of the clinical diagnosis or diagnosis considered by people appears to be less than 50%. Based on these elements, it is impossible to make statistics, especially with such low numbers. In practice, COVID-19 is a disease for which a biological diagnosis is made, not a diagnosis by telephone.

Response: We acknowledge the diagnosis tests that are now available and routinely used to diagnose the infection. Unfortunately, at the time of the study there was a huge lack of diagnostic tests only a reduced number of them were available in specific hospitals. This was the main reason why we included several questions in the survey to address the disease status of the participants. We would like to remark, that these questions were not chosen by us, they were implemented by the health authorities of the Spanish Ministry of Health as diagnostic questions for the general population for imposing quarantine routines. Thus, those questions (can be found in the document of the survey) regarding data on symptoms related to COVID-19 infection allowed us to assign individuals as suspected COVID-19 cases when reporting loss of taste or smell and/or three or more COVID-19 associated symptoms.

Furthermore, the number of people interviewed is extremely low, and finally, there is no control group, as has been done in other, much more convincing studies in the literature, of patients with a comparable pathology and taking another treatment than hydroxychloroquine.

Response: We acknowledge the small number of people included in the study. This is addressed as a limitation factor in the discussion section. This was overcome by the urgency of the situation that forced us to take advantage of the only available source of people receiving this drug chronically as a treatment, the group of patients with autoimmune diseases. 

In addition, the set of untreated cases had to comprise a large number of individuals in order to be able to do the proper matching between treated and untreated individuals, with the treated patients being the limiting group. This matching process would have been difficult to achieve, in a reasonable time, if our control group came only from autoimmune disease patients without treatment.

As we considered that it was a priority to arrive at an answer as quickly as possible, and taking all the available evidence into account, we chose as reference group people from the general population, in order to be able to obtain a representative matchable group that would allow us to answer our question with the maximum guarantees of reliability.

Despite the lack of evidence that autoimmune diseases affect the risk of infection, we cannot completely discard a possible bias caused by the autoimmune status of cases. However, we assume a negligible effect caused by this possible bias in comparison with the other matching parameters such as sex, age and place of residence, that have allowed us to obtain a large and representative N of the different population groups. For these reasons, we consider that the selection of the general population as a reference group is appropriate and allows us to answer, with enough rigor, the main question of this project.

Of course we acknowledge the necessity of clinical trials that will experimentally confirm these results.

In practice, this paper is just a paper of opportunism, which does not contribute to knowledge, and is just being swept away by the flood of COVID-19 and hydroxychlroquine delirium.

Response: Here we must disagree with the reviewer. We did address an important question in a moment of urgency under a pandemic situation in Spain and in many other locations. We used all our available tools to contribute knowledge to COVID-19 disease. We think that our study was properly done and provides epidemiological response to a critical question. 

Reviewer #3: This manuscript describes an interesting study on lack of protective effect of chloroquine derivatives on COVID-19 disease among sample of chronically treated patients in Spain. The manuscript attempts to address an important global public health issue – COVID-19 disease. Understanding treatments that are effective in managing the disease is of paramount importance to policymakers and program managers in addressing this global public health issue. Presently, there is paucity of data on the subject so this manuscript if published could provide some information on the subject.

Response: We appreciate reviewer comments on the aim of the study and the contribution to the COVID-19 disease.

However, my only concern is wrong analysis conducted and the case-control design used in ruling out the protective effect of chloroquine derivatives on COVID-19 disease.

Response: We have addressed all the points raised by the reviewer and we hope that these modifications are enough to consider our manuscript for publication.

See more details below:

Introduction:

a) The authors provided enough background to the study.

Response: Thank you

Methods:

b) The methods employed by the authors in the analysis and presentation of the data appears problematic (see Table 1). The authors appeared to have analysed and presented the results for the treated group against the demographic characteristics separately and also for the untreated group separately. This is inappropriate because the analyses and the presentation of the results must be done across the treated and the untreated groups to be able to identify any protective effect of the intervention in the treated group as against the untreated group. This is a major flaw in this paper. This must be corrected, and more details provided in the methods section about the analysis plans.

Response: We appreciate reviewer comments and we have incorporated changes in order to provide the appropriate tests between treated and untreated groups and to make more comprehensible our method of analysis. 

As pointed, Table 1 shows comparison of demographic characteristic within treatment groups (treated and untreated separately). The major point of this analysis was to demonstrate that there was no difference in the demographic characteristics for COVID-19 positive and negative individuals within each group and thus, that the analysis was not biased due to one of these parameters. We agree that it is also important to show that both treated and untreated groups have also comparable distributions of the demographic characteristics, and thus, that have been properly matched. We have included the comparison between groups for each demographic variable as a new column in Table 1. 

As the reviewer is pointing, the main point of the study is to be able to identify any protective effect of the intervention in the treated group as against the untreated group. This has been addressed by comparing prevalence of COVID-19 between treated and untreated groups (once demonstrated that groups are properly matched). This data is not reported in Table 1 but in the main text and this can make it go unnoticed. The comparison did not reach statistical differences. We also performed the same analysis for “suspected COVID-19 cases”. The P-value for the last comparison was missing and has been included. The text, now in line 160-168, reads as follows:

“The prevalence of declared COVID-19 status in the treated group was 5.3% (95%CI 2.9-7.8) and the mean prevalence among the untreated-matched groups was 3.4% (95%CI 1.4-5.4). Testing differences among treated and 1000 replicates of untreated-matched groups reveals significant differences only in 28 comparisons (P =0.972). In addition, the difference of proportions of declared COVID-19 cases between both groups did not reach statistical significance (difference 1.9%, 95%CI: 0-5.3; P=0.240). Furthermore, the prevalence of suspected COVID-19 patients in treated subjects was of 18.8% (95%CI 14.5-23.1) and the mean prevalence among the untreated-matched groups was 15.7% (95%CI 11.7-19.7). Neither the comparison of the prevalence nor the distribution of the difference of declared COVID-19 cases among groups showed significant differences.”

We have also expanded the materials and methods section for statistical analysis to include the different types of comparisons that are performed in our study. The new text reads as follows (lines 117-128):

“A sample of untreated subjects was matched to the treated group according to sex, age range and incidence region with e1071 R package. The matching process was repeated using a bootstrap strategy and re-sampling of the untreated-matched dataset was repeated 1,000 times to obtain the distribution and mean values of the descriptive statistics such as age range, gender, incidence region and declared COVID-19 prevalence. Statistical analysis was performed to validate the appropriate distribution of the demographic characteristics within each group what rules out the possibility of a bias towards one of the variables. In addition, tests between treated and untreated groups were used to confirm the proper matching of the subjects. Comparisons were performed by Fisher exact test. P-value <0.05 was considered statistically significant. Differences comparison among treated group and 1000 replicates of untreated-matched groups was performed by Fisher exact test. In addition, the difference between the two proportions and a 95% confidence interval for this difference was performed using the comparison of proportions method (Chi-squared test). “

c) Also, randomised studies design could have provided a better alternative to case control design. Thus, absence of randomization could potentially lead to unreliable parameter estimates and its associated misleading conclusion. However, the authors raised genuine concerns in their first round of revision as to why they have used the case control as oppose to other relatively better designs. This is a limitation for the study.

Response: We acknowledge the limitations of our study due to the urgency of the moment and we very much appreciate the reviewer understanding.

d) The selection of controls (untreated) was also problematic, but the authors provided justification for this in their first round of revision, which I consider to be satisfactory.

Response: Thank you

e) Furthermore, the analysis did not adjust for potential confounding among demographic characteristics under consideration in this study because only bivariate analysis via Fisher’s Exact test was conducted. Multivariable analysis is needed to address this concern.

Response: We have included now the analysis within and between groups regarding demographic characteristics. We believe that the existence of no differences in the distribution of these characteristics for any of the comparisons performed is enough to demonstrate the proper matching of the samples and to discard any potential bias due to the different distribution of one of the variables. Thus, we believe that there will not be any extra information added by a multivariable analysis due to the lack of differences between the other variables among treatment groups.

Results:

f) The results appear not to address the research question and the hypothesis postulated. Re-analysis is required. For example, if age range is not statistically significant for the treated group and also not significant for the untreated group, this could not mean that chloroquine or derivatives has no protective effect on COVID-19 disease or vice versa.

Response: We appreciate this comment. We believe that the comparison of COVID-19 prevalence between treated and untreated subjects was unnoticed. We apologise if this is not highlighted enough in the text. However, the analysis to test if chloroquine or derivatives has a protective effect on COVID-19 disease was performed by comparing the prevalence of the disease between treated and untreated subjects that showed no differences.

g) Also, descriptively, you might observe some differences in percentages in COVID-19 prevalence between treated and untreated group BUT statistically, that may not be important (i.e. statistical significance) or vice versa. Hence decision to conclude on any protective effect of chloroquine derivatives on COVID-19 disease should be based on statistical significance based on sound hypothesis testing.

Response: Exactly. This is the test that we have presented, now located in line 160-168. We performed the statistical comparison between COVID-19 distribution between treated and untreated and we showed no statistical differences P=0.972. We have now expanded this section and the methods section to include an extra analysis that reads as follows: 

“The prevalence of declared COVID-19 status in the treated group was 5.3% (95%CI 2.9-7.8) and the mean prevalence among the untreated-matched groups was 3.4% (95%CI 1.4-5.4). Testing differences among treated and 1000 replicates of untreated-matched groups reveals significant differences only in 28 comparisons (P =0.972). In addition, the difference of proportions of declared COVID-19 cases between both groups did not reach statistical significance (difference 1.9%, 95%CI: 0-5.3; P=0.240).”

Discussion and conclusion:

h) The issues raised in the methods and the result sections should be addressed to enable the reviewer do sound evaluation of the discussion and conclusion sections.

Response: We are very thankful to the reviewer for the in deep review of the manuscript. We have addressed all concerns and have modified the required analysis and the corresponding sections in the material and methods, results and discussion. We hope that these modifications are enough to consider our manuscript for publication.

---

## [Decision Letter · Decision Letter 2]

25 Nov 2020

Lack of protective effect of chloroquine derivatives on COVID-19 disease in a Spanish sample of chronically treated patients.

PONE-D-20-26952R2

Dear Dr. Fibla,

We’re pleased to inform you that your manuscript has been judged scientifically suitable for publication and will be formally accepted for publication once it meets all outstanding technical requirements.

Kind regards,

Wenbin Tan

Academic Editor

PLOS ONE

Additional Editor Comments (optional):

Reviewers' comments:

Reviewer's Responses to Questions

**Comments to the Author**

Reviewer #1: All comments have been addressed

---

## [Editor Report · Acceptance letter]

4 Dec 2020

PONE-D-20-26952R2 

Lack of protective effect of chloroquine derivatives on COVID-19 disease in a Spanish sample of chronically treated patients. 

Dear Dr. Fibla:

I'm pleased to inform you that your manuscript has been deemed suitable for publication in PLOS ONE. Congratulations! Your manuscript is now with our production department. 

Kind regards, 

on behalf of

Dr. Wenbin Tan 

Academic Editor

PLOS ONE